# Selected Risk Factors of Developmental Delay in Polish Infants: A Case-Control Study

**DOI:** 10.3390/ijerph15122715

**Published:** 2018-12-02

**Authors:** Marzena Drozd-Dąbrowska, Renata Trusewicz, Maria Ganczak

**Affiliations:** 1Department of Epidemiology and Management, Pomeranian Medical University, 71-210 Szczecin, Poland; mganczak@pum.edu.pl; 2Rehabilitation and Therapy Consulting Room “Corecta”, 71-410 Szczecin, Poland; rentru72@gmail.com

**Keywords:** motor development delay, risk factors, infant

## Abstract

Despite a number of studies on the risk factors of developmental delay (DD) in children conducted in developed countries, Polish data are scarce, which hinder an early diagnosis and initiation of prevention/control measures. Objective: To assess selected risk factors of DD in infants. A case-control survey was conducted in 2017–2018 on 50 infants (≤1 year old) with DD and 104 healthy controls from three outpatient clinics in Szczecin, Poland. Data were collected using an anonymous questionnaire distributed among mothers. The most common risk factors in infants with DD were: Caesarian section (68%), infections (46%), and chronic diseases during pregnancy (48%). DD was significantly correlated with maternal infections and chronic diseases during pregnancy (both: *p* < 0.001), caesarian section (*p* < 0.001), preterm birth (*p* = 0.004), birth weight <2500 g (*p* = 0.03), Apgar score ≤7 (*p* < 0.01), prolonged hyperbilirubinemia (*p* < 0.001), and no breast-feeding (*p* = 0.04). This study reinforces multiple etiologies of DD. Preventive strategies regarding DD in Polish infants should focus on the pre/peri/postnatal risk factors identified in this study. Strategies that prevent and control such risk factors and those on early detection and intervention in high-risk infants are highly recommended.

## 1. Introduction

Child development is influenced by bio-medical and socio-cultural factors that continuously interact [1,2]. An important component of the total development in infancy is motor development, which remains a significant manifestation of functionality and integrity of the central nervous system. Any deviation regarding this type of development can be the first sign of other developmental disorders [3,4].

According to Karsimzadeh, development refers to those variations that a child achieves during life in order to develop physically, mentally, verbally, and socially. Such variations could be affected by numerous factors such as genetic and environmental factors, nutrition, and social stimulants, which in turn may cause a developmental delay (DD) when a child does not achieve developmental milestones within the normal age range [5]. Baker [6] defines DD in children as a term referring to individuals who do not show the expected developmental properties according to their age. This encompasses neurodevelopmental, emotional, and behavioral disorders that have broad and serious adverse impacts on psychological and social well-being.

The tenth revision of the International Statistical Classification of Diseases and Related Health Problems (ICD-10) has four categories of specific DD: Speech and language, scholastic skills, motor function, and mixed specific developmental disorder [7]. According to the fourth edition of the Diagnostic and Statistical Manual of Mental Disorders (DSM-V), specific DD are classified as communication, learning, and motor skills disorders [8]. Children with these disorders require significant additional support from families and educational systems. The disorders frequently persist into adulthood. Notably, those children are at risk for poor academic achievement in the first years of life. This in turn may result in low productivity that leads to low income [2]. 

Infants who are in danger of DD have a medical history of one or more risk factors in the pre, peri, or postnatal period. The risk factors for DD include maternal and infant biological, psychosocial (individual and familial), and environmental factors. In the period before birth, such factors as young maternal age [9,10,11], short interval between pregnancies, history of previous abortion [11], multiple gestation [1,4], preeclampsia, placental abruption, immaturity and intrauterine growth restriction [11], a mother’s underlying diseases (including multi-morbidity [4,12] and addiction), being deprived of primary care during pregnancy, low maternal educational level [2,9], and being a single mother household [2,4] are considered to increase the risk of DD in the infant. Delivery through caesarian section [11] and preterm birth [1,13,14,15,16] are the most important risk factors of DD in the perinatal period, and male gender [4,9,13], low birth weight [1,2,4], first minute Apgar score <7 [11], intracranial hemorrhage [17], kernicterus, and no breastfeeding [1] are all risk factors in the postnatal period.

Worldwide, the prevalence of DD in children is reported to be about 12–18% [2,18], however, in high risk infants, it is even higher. As an example, motor DD in this high-risk group is observed about 30% more than in the general population [4]. Therefore, early detection of infants with developmental disorders should be performed at an early age [1,2,3,4]. Previous reports have shown that early intervention programs are cost-effective and have life-long benefits and optimal developmental accomplishment [1,2,19].

The Polish health care system provides free medical services for all infants, and almost all parents will make multiple visits (every two–three months) to family medical clinics for immuni-zation, weight control, and developmental check-ups of their children [20]. Just as in some other EU countries [13], the latter is usually carried out by a general practitioner (GP) and based on clinical judgement together with the mother’s medical history rather than with the use of standardized screening or assessment tools. If any early diagnosis and intervention at the primary care level is needed, the local GP refers an infant to specialist services, such as neurologists, orthopedic surgeons, and physiotherapists. However, without standardized instruments, it is difficult for GPs to adequately detect DD in infants. 

An essential first step, not only to make an early diagnosis but also to initiate prevention and control measures in terms of reducing the burden of DD, is knowledge of its risk factors, which may differ in each society [18]. Despite a number of studies on the prevalence of DD and contributing risk factors in developed countries, Polish data are rather scarce and there is a pressing need for relevant Polish reports on the subject. Considering the critical significance of addressing the most common risk factors in infants with DD and the limited research from Poland in this area, the present case-control study was designed to assess selected pre, peri, and postnatal risk factors in Polish infants aged 0–12 months with diagnosed DD. This in turn can provide the possibility to create mother and infant-friendly environments that support optimal motor development. 

## 2. Materials and Methods

### 2.1. Design and Setting

The case-control study was conducted from November 2017 to March 2018 in Szczecin, Poland, among 0–12 month old infants.

### 2.2. Study Population

The studied population accounted for 154 children, 50 cases and 104 controls up to 12 month old. The children diagnosed with DD (cases) were compared with healthy children (controls) with respect to pre, peri, and postnatal risk factors. The case definition was: Being up to 12 months old and diagnosed with DD by a neurologist and/or orthopedic surgeon. Children were diagnosed with DD by the specialists after the initial assessment made by GPs from primary care clinics in the Szczecin region. Cases were selected from children with diagnosed DD that were referred to two rehabilitation and therapy outpatient clinics in Szczecin for neurodevelopmental therapy. Developmental evaluation included five motor development areas (gross and fine motor skills), cognitive and emotional development, and communication (perception and speech) development. The control group was selected from the same population from which the cases had come. They were recruited by GPs during qualifications for preventive vaccinations in one primary care clinic in Szczecin. Children with congenital anomalies were excluded from the study.

### 2.3. Study Instrument

Data were collected using an anonymous questionnaire with 31 questions for mothers of the abovementioned infants about selected risk factors of DD. It consisted of two parts: 1. Socio-demographic data including maternal age, place of residence, education level, socio-economic status, and the number of children in the family; 2. the prenatal, perinatal and postnatal history (prenatal risk factors: Infections and chronic diseases during pregnancy, multiple gestation and habitual abortions; perinatal risk factors: Time and the type of delivery; postnatal risk factors: Weight of the newborn, Apgar score in the first minute after delivery, hyperbilirubinemia, breastfeeding).

Mothers of infants were included in the study after obtaining written signed informed consent. At the Pomeranian Medical University, there is no requirement for ethics committee approval for anonymous case-control studies that use questionnaires. Nevertheless, before completing a questionnaire, a written explanation of the objectives of the research was given to all mothers who were then assured that they would not be identified in any presentation or publication. They were also assured that their participation would be on a voluntary basis, as well as that they had full rights to withdraw from the study at any time. To protect the confidentiality of the subjects, completed questionnaires were stored in a locked filing cabinet and computer data were password protected and accessible only to the three study investigators.

### 2.4. Statistical Analysis

Data analysis was carried out using STATISTICA PL, Version 12.5 (StatSoft, Kraków, Poland, 2016). Frequencies and percentages were used for categorical variables to describe the characteristics of infants and mothers, and continuous variables were reported as mean ± standard deviation. The categorical variables were compared (bivariate analyses) using the Pearson’s chi-square test. Associations of variables with outcomes were expressed by odds ratio (OR) with 95% confidence interval (95% CI). Statistical significance was set at *p* < 0.05.

## 3. Results

### 3.1. Socio-Demographic Characteristics of the Case and the Control Groups

The response rate was 100%. Table 1 demonstrates the socio-demographic characteristics of the case and the control groups. There were no significant differences between groups in terms of maternal age and education level, presence of more than one child in the family, and socio-economic status of the family. However, a statistically significant difference was found between the case and the control group in terms of place of residence.

### 3.2. Frequency of DD Risk Factors 

The most common risk factors in infants with DD were: Caesarian section (68%), infections and chronic diseases during pregnancy (46% and 48%, respectively), followed by severe neonatal hyperbilirubinemia (28%), prematurity (20%), and low birth weight (20%), as shown in Table 2.

### 3.3. Differences Regarding the Selected Risk Factors of DD

Between-group differences regarding the selected risk factors of DD are shown in Table 2. When taking into accounts prenatal risk factors, there were significant between-group differences observed regarding maternal chronic diseases (*p* < 0.001) and infections during pregnancy (*p* < 0.001). The most common infections during pregnancy reported by mothers of infants with DD were respiratory (24%) and urinary tract infections (14%), while in controls only 2.9% of mothers reported a respiratory tract infection and no other infections were reported.

Regarding perinatal risk factors, there were significant differences between cases and controls concerning preterm birth (*p* = 0.004) and caesarian section (*p* < 0.001). Statistically significant between-group differences regarding postnatal risk factors were observed regarding the first minute Apgar score (*p* = 0.01), low birth weight (*p* = 0.03), prolonged hyperbilirubinemia (*p* < 0.001), and breastfeeding (*p* = 0.04).

## 4. Discussion

### 4.1. Results Overview

The most common risk factors in infants with DD were mother-related (chronic diseases and infections during pregnancy) as well as medical care-related (caesarian section). DD was significantly associated with pre, peri, and postnatal factors such as: Habitual abortions, maternal systemic infection during pregnancy, maternal chronic disease, caesarian section, prematurity, neonatal hyperbilirubinemia, hospitalization at the ICU, and no breastfeeding.

### 4.2. Factors Associated with Developmental Disorders

The main aim of this study is to build a more comprehensive picture of infants’ DD by determining its predictors. A high-risk pregnancy has a significant correlation with DD in infants [19,21]. This was also confirmed by the results of this study. Statistically significant differences between cases and controls were found for numerous prenatal risk factors, such as chronic diseases during pregnancy and infections. The latter was associated with 29 times higher odds of DD, however, this is a modifiable risk factor. It is noteworthy that compared with nonpregnant women, pregnant women are more severely affected by infections with numerous organisms, including the influenza virus, hepatitis E, herpes simplex, measles, smallpox, and varicella virus. This is due to immunologic alterations with advancing pregnancy that may impair pathogen clearance, resulting in an increased severity of disease caused by some pathogens [22]. Therefore, vaccination before and during pregnancy, which has proved safe and effective for a number of infectious agents, is strongly recommended. The beneficial effects of maternal vaccination may not only be limited to the mother but, by reducing fetal inflammation, may also provide long-term benefits to the child. The education of pregnant women on the prevention of infections and the early identification and appropriate treatment of infectious diseases during pregnancy remain important strategies for protecting infants regarding DD.

Other prenatal factors, such as teratogenic drugs, radiation, and vaginal bleeding, were also highlighted as risk factors of infants’ DD through their contribution in causing asphyxia and injuries to the developing brain [23]. 

In this study, the frequency of perinatal risk factors, such as preterm birth and caesarian section, were significantly higher in the group of cases compared to controls. The preterm birth (≤37 gestational week) was related with a six times higher risk of DD in this study and this was also confirmed by others [1,24]. As an example, Stoelhorst investigated the effect of prematurity on developmental outcomes and concluded that, at 18 and 24-months corrected age, 40% of very prematurely born infants suffered from delayed mental development, psychomotor development, or both [25]. 

According to the study results, infants born by caesarean section were nine times more likely to present DD later in life compared to infants born by vaginal delivery, also confirmed by others [26]. The findings underline the need for a precautionary approach in responding to requests for a planned cesarean when there are no apparent elevated risks from vaginal birth [27]. Further studies to test these findings on a larger cohort of children and examine whether the correlation exists in an older age group would be of value.

Regarding postnatal risk factors, a lower Apgar score in the first minute was linked to significantly higher risk of DD. Our results support previous evidence related to this issue [28]. However, some studies reported a lack of correlation between the Apgar score and further DD in children [24]. Additional studies at regional or national level are needed to better assess this issue.

Low birth weight may result from a shorter pregnancy, intrauterine delay, or a combination of both [1]. The analysis in the subgroups of low and normal birth weight infants indicated that those with birth weight below 2500 g were three times more likely to develop DD. Our results are coherent with previous findings that reported a marked reduction in the suspicion of DD with an increase in child birth weight [1,24]. As an example, children with moderate low birth weight (1500–2499 g) achieved significantly lower scores in fine motor skills in comparison to those with normal birth weight (2500–4000 g) [29]. Although not assessed in our study, a similar effect was found for several other indicators (gestational age, cephalic perimeter, and length) [30]. 

Infants with prolonged hyperbilirubinemia were six times more likely to develop DD compared to the control group. Similarly, in a study carried out in Tehran on 200 cases with DD, severe hyperbilirubinemia played a detrimental role in causing DD [31]. 

Our findings indicated that breastfeeding was related to lower risk of DD in surveyed children, with similar results being found by others. Breastfeeding was reported as an independent effect in relation to the developmental status at the 12th month of life [32]. Children who were never breastfed had an 88% increase in the chance for suspected DD compared to those who were breastfed for more than six months in a study conducted by Halpern [1]. A more recent study found that infants who had never been breastfed were 50% more likely to have gross motor coordination delays than infants who had been breastfed exclusively for at least four months [33]. Hence, health workers should support pregnant women by establishing and sustaining appropriate breastfeeding practices after delivery and provide further skilled support to breastfeeding mothers.

No significant relationships between DD of children and maternal age, place of residence, socio-economic status of family (SES), maternal education, and the number of children in family were found in this study. Similarly, Valla et al. in a study of DD in Norwegian infants between 4 and 12 months did not find any significant association between maternal education level and the suspected DD [13].

Our results might be influenced by selection bias. Poorer parents could have more often resigned from their child’s therapy and were therefore not included in the survey. The low participation rate regarding mothers with poor SES might be related to the financial restrictions due to, e.g., the distance to cover to reach the health care facility and transportation costs. In addition, the questionnaire queried the mothers about self-assessment of family SES. This could be erroneously estimated. 

Most studies confirm the existence of an association between SES and DD in children [1,30,34,35,36,37]. As an example, Potjik et al. found mother’s lower education level, lower income, and poor housing conditions significantly correlated with child’s DD [34]. A decrease in family SES significantly increased the frequency of DD in Iranian children surveyed by Ahmadi Doulabi et al. [38]. The probability of being suspected of DD was found to be twice as high in children of lower income families than when compared to those parents with higher income [30] and twice as high in children with more than three siblings. Regarding maternal schooling, the risk increased as maternal schooling decreased [1].

### 4.3. Limitations

This study has a number of limitations. Firstly, the number of cases was relatively small and the study was only conducted in one province. Therefore, the findings do not necessarily apply to all Polish children. Further studies at a national level would be of value. Secondly, response rates concerning some DD risk factors were relatively low, possibly due to a recall bias [39], i.e., mothers who had a child with DD tried to identify some risk factors during pregnancy or childbirth more frequently than when compared to those of the healthy children. A bias associated with self-report questionnaires is quite common and can potentially influence the outcome. Therefore, despite the strong relationships between DD and some risk factors identified in this study, the findings should be interpreted with caution. There might also be a selection bias in this study. The cases were selected from rehabilitation and therapy consulting rooms whereas controls were from the primary care clinic. Still, no statistically significant differences in socio-demographic characteristics between case and control groups were found, except the place of residence. While we highlighted numerous variables that referred to DD, other determinants might have also influenced the DD in surveyed infants.

### 4.4. Implications for Mother and Child Care

The most common risk factors identified in the surveyed infants were chronic diseases and infections during pregnancy, as well as caesarian section. As previous studies showed that the accumulation of risk factors determines a higher impact on child DD [1,30], preventive strategies regarding Polish infants should focus on minimizing the pre, peri, and postnatal DD risk factors identified in this study.

The abovementioned risk factors are mother-dependent, but also medical care-dependent. As such, mother-dependent factors like infections and chronic diseases during pregnancy, preterm birth, and caesarian section should be of special interest to gynecologists and obstetricians in planning individualized prophylaxis, treatment, and care oriented to pregnant women. Other physicians, particularly pediatric and family medicine practitioners, should pay special attention regarding high-risk infants identified in this study, specifically those with low birth weight, low Apgar score, prolonged hyperbilirubinemia, and hospitalization at the ICU, and systematically monitor their developmental status. A leading role of those practitioners is evitable. Evidence shows that early detection of infants with DD risk factors could improve the successful functioning of the affected children. [15,19,20,40,41].

## 5. Conclusions

The present study assessed selected risk factors of DD in infants with the results mostly supporting previous findings from other countries, both in the European region and outside of Europe, and reinforcing the multiple etiology of DD [1,2,4,9,15,18,19,27,32]. Preventive strategies regarding DD in Polish infants should focus on the pre-, peri- and postnatal risk factors identified in this study. Our results show that high-risk pregnancy, specifically maternal infections and chronic diseases, were associated with the highest risk of DD. Therefore, strategies that control and prevent such risk factors are highly recommended. Planning individualized prophylaxis, treatment, and care oriented to pregnant women could prevent DD in infants. Comprehensive mother and child care, which helps to achieve and maintain good physical and mental health in newborns and infants, would have a beneficial influence on the reduction of DD incidence in early infancy. For infants with recognized DD risk factors, early detection and intervention could reduce its symptoms.

## Figures and Tables

**Table 1 ijerph-15-02715-t001:** Socio-demographic characteristics of the study groups; Poland, 2018.

Variable	Cases *N* = 50	Controls *N* = 104	*p*
	*n* (%)	*n* (%)	
*Maternal age*			0.85
20–30 years	21 (42.0%)	42 (40.4%)	
31–40 years	29 (58.0%)	62 (59.6%)	
*Infant residence*			<0.001 *
City ≤ 400,000 inhabitants	15 (30.0%)	1 (1.0%)	
City >400,000 inhabitants	35 (70.0%)	103 (99.0%)	
*Socio-economic status of family*			0.99
High	11 (22.0%)	22 (21.2%)	
Medium/low	39 (78.0%)	80 (78.8%)	
*Maternal educational status*			0.22
Below high	5 (10.0%)	20 (19.2%)	
High	45 (90.0%)	84 (80.8%)	
*Number of children in family*			0.12
1	24 (48.0%)	33 (31,7%)	
>1	26 (52.0%)	71 (68.3%)	

* *p* ≤ 0.05.

**Table 2 ijerph-15-02715-t002:** Analysis of between-group differences regarding selected risk factors of developmental delay (DD); Poland, 2018.

Variable	*N*		*N*			OR	
	Cases	%	Control	%	OR	95%CI	*p*
Prenatal risk factors							
*Maternal age*							
20–30 years	29	42	42	40.4	1	0.38; 1.37	0.86
>30 years	31	58	62	59.6	0.72		
*Multiple gestation*							
No	44	88	90	86.5	1	0.32; 2.44	0.06
Yes	6	12	14	13.5	0.88		
*Habitual abortions*							
No	41	82	95	91.4	1	0.94; 7.24	0.06
Yes	9	18	8	7.7	2.61		
No data	0	0	1	0.9			
*Infection during pregnancy*							
No	27	54	101	97.1	1	8.01; 102.8	<0.001 *
Yes	23	46	3	2.9	28.7		
*Chronic disease during pregnancy*							
No	26	52	96	94.1	1	5.47; 39.9	<0.001 *
Yes	24	48	6	5.9	14.77		
Perinatal risk factors							
*Preterm birth*							
No	29	58	75	72.1	1		0.004 *
Yes	10	20	4	3.8	6.46	1.88; 22.2	
No data	11	22	25	24.1			
*Caesarian section*							
No	16	32	84	80.8	1		<0.001 *
Yes	34	68	19	18.3	9.39	4.32; 20.39	
No data	0	0	1	0.9			
Postnatal risk factors							
*Birth weight*							
≥ 2500g	40	80	96	92.3	1		0.03 *
<2500g	10	20	8	7.7	3	1.10; 8.16	
*Apgar score*							
10–8	43	86	103	97	1		
0–7	4	8	1	3	9.58	1.04; 88.21	0.01 *
No data	3	6	0	100			
*Prolonged hyperbilirubinemia*							
No	34	68	97	93.3	1		
Yes	14	28	7	6.7	5.7	2.12; 15.3	<0.001 *
No data	2	4	0	0			
*Breast feeding*							
No	19	38	23	22.1	1	0.22; 0.96	0.04 *
Yes	31	62	81	77.9	0.46		

* *p* ≤ 0.05.

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
