# Peer review of "Selected Risk Factors of Developmental Delay in Polish Infants: A Case-Control Study"

_ijerph, 2018, doi:10.3390/ijerph15122715_

Round 1

Reviewer 1 Report

Title: Risk factors of developmental delay in Polish infants: a case-control
study
Authors: Marzena Drozd-Dąbrowska *, Renata Trusewicz, Maria Ganczak

Introduction

This is generally well written and informative. The authors have given a good explanation of the issues surrounding Developmental Disorders that are included in the study. However, I think it would be useful to refer to DSM-5 (or even ICD-10) as it is currently the manual that is consulted for such disorders.

Page 1 Line 38 to 42 I realise that you have given references at the end of this section but it would be really helpful if you could reference each factor in turn.

Page 2 Line 49 Intervention (not inter-vention)

Method

Participants

Could you detail which participants were in each group as currently it isn’t clear.

Study Instruments

Page 2 Line 81It should be Data were (data is plural)

Page 2 Line 81 with 31 questions for mothers of . . .

Page 2 Line 87 hospitalization in (not at)

Results

This section is well written and reports the findings well.

Discussion

This section has been constructed and discussed well.  Relevant literature has been used to compare and contrast the findings of this study;

Page 5 Line 138 hospitalization in (not at)

Page 7 Line 198 (sacker) should this be number 31?

Page 7 Line 235 hospitalization in (not at)

Conclusion

The conclusion is well written and makes some recommendations for strategies that control and prevent such risk factors; this is important if the number of infants that develop DD is to be reduced.

References

All appear to be fine.

Figures and Tables

All appear to be fine.

Author Response

Response to the comments of the Reviewer 1:

I.                     Introduction:

This is generally well written and informative.      The authors have given a good explanation of the issues surrounding      Developmental Disorders that are included in the study. However, I think      it would be useful to refer to DSM-5 (or even ICD-10) as it is currently      the manual that is consulted for such disorders.  

Response 1:

According to the Reviewer’s comment, we referred to DSM-5 and ICD-10 as follows:

“The tenth revision of the International Statistical Classification of Diseases and Related Health Problems (ICD-10) has four categories of specific DD: speech and language, scholastic skills, motor function, and mixed specific developmental disorder [7]. According to the fourth edition of the Diagnostic and Statistical Manual of Mental Disorders (DSM-V), specific DD are classified as communication, learning and motor skills disorders [8]. Children with these disorders require significant additional support from families and educational systems; the disorders frequently persist into adulthood. Notably, those children are at risk for poor academic achievement in the first years of life. This in turn may result in low productivity that leads to low income [2].

Page 1 Line 38 to 42 I realise that you have given references at the end of this section but it would be really helpful if you could reference each factor in turn.

According to the Reviewer’s suggestion this has been corrected as follows:

“In the period before birth, such factors as young maternal age [9,10,11], short interval between pregnancies, history of previous abortion [11], multiple gestation [1,4], preeclampsia, placental abruption, immaturity and intrauterine growth restriction [11], a mother's underlying diseases, including multi-morbidity [4,12], addiction, and being deprived of primary care during pregnancy, but also low maternal educational level [2,9] and single mother household [2,4], are considered to increase the risk of DD in the infant. Delivery through caesarian section [11] and preterm birth [1,13,14,15,16] are the most important risk factors of DD in the perinatal period, while male gender [4,9,13], low birth weight [1,2,4], first minute Apgar score <7 [11], intracranial hemorrhage [17], kernicterus, as well as no breastfeeding [1] – are all risk factors in the postnatal period.”

Page 2 Line 49 Intervention (not inter-vention)”

Response 2:

According to the Reviewer’s comment this has been corrected:

 “performed at an early age [1-4]. Previous reports have shown that early intervention programs are cost-effective”

II.                   Method:

Participants

Could you detail which participants were in each      group as currently it isn’t clear.

Response 1:

According to the Reviewer’s comments, we clarified this issue as follows:

2.2. Study population

“The studied population accounted for 154 children, 50 cases and 104 controls up to 12 month old. The children diagnosed with DD (cases) were compared with healthy children (controls) with respect to pre-, peri- and postnatal risk factors. The case definition was: being up to 12 months old, diagnosed with DD by a neurologist and/or orthopedic surgeon. Children were diagnosed with DD by the specialists after the initial assessment made by GP’s from primary care clinics in Szczecin region. Cases were selected from children with diagnosed DD referred to two rehabilitation and therapy outpatient clinics in Szczecin for neurodevelopmental therapy. Developmental evaluation included five motor development areas (gross and fine motor skills), cognitive and emotional development, communication (perception and speech) development. The control group was selected from the same population from which the cases have come. They were recruited by GP’s during qualifications for preventive vaccinations in one primary care clinic in Szczecin. Children with congenital anomalies were excluded from the study.”

 Study Instruments

Page 2 Line 81It should be Data were (data      is plural)

Page 2 Line 81 with 31 questions for mothers of . . .

Response 1:

According to the Reviewer’s comments, this has been corrected as follows:

“Data were collected using an anonymous questionnaire with 31 questions for mothers of abovementioned infants about selected risk factors of DD.”

Page 2 Line 87      hospitalization in (not at)

Response 2:

According to a Reviewers suggestion we have decided to delete the variable “hospitalization in ICU” from our analysis, so this sentence is written as follows:

”the newborn, Apgar score in the first minute after delivery, hyperbilirubinemia and breastfeeding”

III.                 Discussion:

Page 5 Line 138 hospitalization in (not      at)

Page 7 Line 235 hospitalization in (not at)

Response 1:

According to a Reviewer’s suggestion we have decided to delete the variable “hospitalization in ICU” from our analysis.

Page 7 Line 198 (sacker) should this be number      31?

Response 2:

According to the Reviewer’s comment, we put [33] in the bracket to number the reference by Sacker et al.

“A more recent study found that infants who had never been breastfed were 50% more likely to have gross motor coordination delays than infants who had been breastfed exclusively for at least 4 months [33].”

Reviewer 2 Report

The current study addresses the main risk factors that might impact on developmental delay in Polish neonates and infants. The study is appropriately designed and is based on a representative sample. The identified factors are sound. My main concern applies to hospitalization at a neonatal intensive care unit as a factor. Since all the other factors selected for the study are primarily of a causal nature, this latter one does not seem to belong here. being hospitalised at the ICU may be simply the result of premature birth or any of the other risk factors. the authors should explain exactly what role they attribute to this factor and comment on its nature. Since such studies are supposed to be informative, they should treat the factors they identify at the appropriate level - hospitalization at a neonatal ICU is not a causal factor and should be properly described (what is its nature etc.). Furthermore, this is mentioned in line 229 as a "medical care-dependent" factor. Do they mean that something has happened to the neonates/infants while in medical care??

The authors write ".. referring to individuals who do not show the expected developmental properties according to their age" (lines 32-33). This should be re-phrased according to scientific terminology in the field.

The SES factors (lines 205-208) should be commented/discussed and supported by other research in the field.

Author Response

Response to the comments of the Reviewer 2:

The      current study addresses the main risk factors that might impact on      developmental delay in Polish neonates and infants. The study is      appropriately designed and is based on a representative sample. The identified      factors are sound.

My main concern applies to hospitalization at a neonatal intensive care unit as a factor. Since all the other factors selected for the study are primarily of a causal nature, this latter one does not seem to belong here. being hospitalised at the ICU may be simply the result of premature birth or any of the other risk factors. the authors should explain exactly what role they attribute to this factor and comment on its nature. Since such studies are supposed to be informative, they should treat the factors they identify at the appropriate level - hospitalization at a neonatal ICU is not a causal factor and should be properly described (what is its nature etc.).”

Response 1:

We do agree with the Reviewer that being hospitalized may be simply the result of numerous risk factors and is not a casual factor. However, findings from the literature suggest that neonatal ICU graduates are most at risk for DD (Lehner DC at al.). Therefore, we thought it would be an important informative message for primary care practitioners to further observe this group of children and – in the case of any particular need – to use formal developmental screening tests and parent report surveys of the infants to minimize the risk of delays. According to the Reviewer’s suggestion we have decided to delete this variable from our analysis, as well the relevant paragraph from the Discussion section.

Furthermore,      this is mentioned in line 229 as a "medical care-dependent"      factor. Do they mean that something has happened to the neonates/infants      while in medical care?  

Response 2:

In line 229 we refer to the medical care dependent factors. Being hospitalized on ICU could be one of those factors. However, there are also some other medical care-dependent factors possibly influencing DD in the surveyed children, such as preterm birth and caesarian section. Therefore, despite deleting “hospitalization in neonatal ICU” as a risk factor, we have decided to retain the sentence “The abovementioned risk factors are mother-dependent, but also medical care-dependent.”

“The authors      write ".. referring to individuals who do not show the expected      developmental properties according to their age" (lines 32-33). This      should be re-phrased according to scientific terminology in the field.”

Response 3:

We would like to thank the Reviewer for this valuable comment. To rephrase the previous statement, according to scientific terminology in the field, we added following fragment:

“According to Karsimzadeh development refers to those variations that a child achieves during life in order to develop physically, mentally, verbally and socially. Such variations could be affected by numerous factors, such as genetic, as well as environmental factors, nutrition and social stimulants which in turn may cause a developmental delay (DD), when a child does not achieve developmental milestones within the normal age range [5]. Baker [6] defines DD in children as a term referring to individuals who do not show the expected developmental properties according to their age. This encompasses neurodevelopmental, emotional, and behavioral disorders that have broad and serious adverse impacts on psychological and social well-being.

The tenth revision of the International Statistical Classification of Diseases and Related Health Problems (ICD-10) has four categories of specific DD: speech and language, scholastic skills, motor function, and mixed specific developmental disorder [7]. According to the fourth edition of the Diagnostic and Statistical Manual of Mental Disorders (DSM-V), specific DD are classified as communication, learning and motor skills disorders [8]. Children with these disorders require significant additional support from families and educational systems; the disorders frequently persist into adulthood. Notably, those children are at risk for poor academic achievement in the first years of life. This in turn may result in low productivity that leads to low income [2].”

“The SES factors      (lines 205-208) should be commented/discussed and supported by other      research in the field.”

Response 4:

“No significant relationships between DD of children and maternal age, place of residence, socio-economic status of family (SES), maternal education and the number of children in family were found in this study. Similarly, Valla et al. in a study of DD in Norwegian infants between 4 and 12 months did not find any significant association between maternal education level and the suspected DD [13].

Our results might be influenced by selection bias; poorer parents could have more often resigned from their child’s therapy and therefore were not included in the survey. The low participation rate regarding mothers with poor SES might be related to the finance restrictions due to e.g. the distance to cover to reach the health care facility and transportation costs. In addition, the questionnaire queried the mothers about self-assessment of family SES, this could be erroneously estimated.

Most studies confirm the existence of an association between SES and DD in children [1,30,34,35,36, 37]. As an example, Potjik et al. found mother’s lower education level, lower income and poor housing conditions significantly correlated with child’s DD [34]. A decrease in family SES significantly increased the frequency of DD in Iranian children surveyed by Ahmadi Doulabi et al. [38]. The probability of being suspected of DD was found to be twice as high in children of lower income families than when compared to those parents with higher income [30], and twice as high in children with more than three siblings. Regarding maternal schooling, the risk increased as maternal schooling decreased [1]. “

Reviewer 3 Report

The manuscript „Risk factors of developmental delay in Polish infants: a case-control study” by M.Drozd-Dabrowska, R. Trusewicz and M. Gonczak evaluates different factors which correlate with or may contribute to the developmental delay (DD) in Polish infants. The study was based on case-control survey of mothers.  Fifty infants previously diagnosed with DD were compared to 104 presumably healthy infants – all from the region of Szczecin.     

The results show that caesarian section, hospitalization on ICU, multiple gestations, infectious and chronic diseases during pregnancy were the dominant factors linked to DD.  Other factors, which apparently contributed less, were: preterm birth, low birth weight, low Apgar score, prolonged hyperbilirubinemia, and lack of breast feeding.    

The paper provides potentially valuable information, but in the present form it has several shortcomings which significantly diminish its qualify.

The results and methods are presented in a sloppy way and sometimes they are inaccurate.  The list of specific questions and problems:

Study design: Was the study approved by IRB? Please clarify.

Line 69:  it is written “… the study was conducted from November to March 2018”.  What does it mean? From March to November 2018 or from November 2017 to March 2018? 

The complete list of 31 questions querying the mothers about risk factors for DD needs to be included.  Was there a question about vaccinations and vaccine reactions? Such question is essential, as the European and American databases show hundreds of thousands of adverse vaccine reactions, many of which require hospitalizations and end up with death. Such adverse vaccine reactions may have contributed to DD in children. These issues need to be judiciously examined and elaborated.  Without them the manuscript is markedly inadequate.   

Investigate the ages and medical reasons for infant hospitalizations and their potential chronological links to vaccinations, which start on day 1 of life and are repeated every 2-3 months with mixtures of different vaccines.  In countries such as Poland, where vaccines are given to newborns during first hours after birth, many infants react with collapse, develop seizures and other life-threatening symptoms, hence they must be reanimated and hospitalized for days or weeks.  As several studies document, neonatal vaccinations contribute to DD in children and discussing this issue is critical.   

Were there questions about alcohol/drug use during pregnancy, about home violence and inadequate nutrition included?  These factors are known to contribute to DD.               

Results: The numbers seem to be wrongly placed in the row: “Multiple gestation” and the OR seems wrong, showing reduced risk of DD for multiple gestation group (OR=0.39) – inconsistent with data description.     

Fig 1 is a graphic presentation of data from Table 2, it may be redundant. It also contains errors.  The bars showing chronic disease and infection during pregnancy do not correspond to data in Table 2.

Discussion:  In a small population of cases (only 50 mothers of children with DD were queried)  many risk factors overlap. E.g. 50 % of women had multiple gestations and nearly 50% had infections or chronic disease during pregnancy.   Were these risk factors occurring in the same women?    If yes, they may be linked and this would influence the interpretation of data.

It has been known for centuries that infections during pregnancy are risk factors for DD, but advocating immunizations of pregnant women is erroneous, in my opinion.  Wise, experienced obstetricians always promoted minimal or no medical interventions in healthy pregnant women.  Immunizations introduce many toxic substances (Al, Hg, formaldehyde, detergents, antibiotics, viruses, bacterial toxins, foreign genetic materials etc.) to blood of mothers and fetuses.  These substances and vaccines have never been tested for safety and efficacy in pregnant women, and cannot be beneficial for fetal development. Neither they are safe for pregnant mothers.  The evidence for this comes from US national statistics. The US is one of a few countries in the world where many pregnant women are injected with vaccines. And the results of this practice are devastating: there are high numbers of fetal deaths/losses in vaccinated mothers, US has the highest infant and maternal mortality rates among all developed counties, and has large numbers of brain-injured children.  On the other hand, Norway, which does not promote maternal immunizations, has a low percentage of children with DD (about 5%).  Healthy diet and cognizant avoidance of infections during pregnancy is a prudent way of protecting the fetus. 

Author Response

Response to the comments of the Reviewer 3:

“The manuscript „Risk factors of developmental delay in Polish infants: a case-control study” by M.Drozd-Dabrowska, R. Trusewicz and M. Ganczak evaluates different factors which correlate with or may contribute to the developmental delay (DD) in Polish infants. The study was based on case-control survey of mothers.  Fifty infants previously diagnosed with DD were compared to 104 presumably healthy infants – all from the region of Szczecin.     

The results show that caesarian section, hospitalization on ICU, multiple gestations, infectious and chronic diseases during pregnancy were the dominant factors linked to DD.  Other factors, which apparently contributed less, were: preterm birth, low birth weight, low Apgar score, prolonged hyperbilirubinemia, and lack of breast feeding.    

The paper provides potentially valuable information, but in the present form it has several shortcomings which significantly diminish its qualify.

The results and methods are presented in a sloppy way and sometimes they are inaccurate.  The list of specific questions and problems:

I.                    Study design:

1.Was the study approved by IRB? Please clarify.

Response 1:

“At the Pomeranian Medical University, there is no requirement for ethics committee approval for anonymous case-control studies which use questionnaires. Nevertheless, before fulfilling a questionnaire, a written explanation of the objectives of the research was given to all mothers, who were then assured that they would not be identified in any presentation or publication. They were also assured that their participation would be on a voluntary basis, as well as that they had full rights to withdraw from the study at any time. To protect the confidentiality of the subjects, completed questionnaires were stored in a locked filing cabinet, and computer data were password protected and only accessible by the three study investigators.”

According to the Reviewer’s suggestion this explanation has been included to the study at the end of the Methods section.

2. Line 69:  it is written “… the study was conducted from November to March 2018”.  What does it mean? From March to November 2018 or from November 2017 to March 2018? 

Response 2:

According to the Reviewer’s suggestion this has been corrected as follows:

 “The case-control study was conducted from November 2017 to March 2018 in Szczecin, Poland, among 0-12 months old infants.”

3. The complete list of 31 questions querying the mothers about risk factors for DD needs to be included.  

Response 3:

A researcher-administered anonymous questionnaire that queried mothers about the risk factors for DD was designed by the study team using a literature review. It queried the mothers about socio-demographic characteristics, as well as selected risk factors of DD. All of those are listed in Tables 1 and 2. Therefore, in the opinion of the authors there is no need to include the questionnaire in the manuscript. However, the questionnaire may be available for potential readers on request through the dedicated e-mail address.

4. Was there a question about vaccinations and vaccine reactions?

Response 4:

There were three questions regarding vaccinations in infants in the study questionnaire which are as follows:

Has a child been vaccinated according to the      National Immunization Program (yes/no/don’t remember)

Has a child been vaccinated with additional      self-paid vaccines? (yes/no/don’t remember)

If a child has been vaccinated with an additional      self-paid vaccine, which one was it: rotavirus, varicella, influenza,      meningococcal disease?

There was no question about vaccine reactions (side effects). In Poland all side effects related to vaccinations have to be registered in the national database. According to this registry, there were 68 cases reported in the West Pomeranian region (in which the study was conducted) in 2017. Twelve of those were classified as serious according to the WHO definition. Because serious side effects after vaccinations, which might have contributed to DD children, are extremely rare in Poland, it could be assumed that this did not influence the study results.

5. Were there questions about alcohol/drug use during pregnancy, about home violence and inadequate nutrition included?  These factors are known to contribute to DD.       

Response 5:

Indeed, we have not asked the mothers about alcohol/drug use during pregnancy/home violence and inadequate nutrition. These factors are known to contribute to DD. This is explicitly stated in the study objective as follows:

“…the present case-control study was designed to assess selected pre-, peri- and postnatal risk factors in Polish infants aged 0-12 months with diagnosed DD.”

According to the Reviewer’s suggestion, we added “selected” to the tittle of our manuscript as follows:

“Selected risk factors of developmental delay in Polish infants: a case-control study”

                                    II. Results:

1.The numbers seem to be wrongly placed in the row: “Multiple gestation” and the OR seems wrong, showing reduced risk of DD for multiple gestation group (OR=0.39) – inconsistent with data description.     

Response 1:

We would like to thank the Reviewer for this valuable comment. Indeed, the numbers were wrongly placed in the row. We have corrected them adequately to those in the database. These are follows:

Multiple gestation - No: N cases=44; 88%, N cotrols=90; 86.5% OR=1.0

Multiple gestation - Yes: N cases=6; 12%, N cotrols=14; 13.5% OR=0.88 95%CI [0.32;2.44]

p=0.06

We are sorry for the mistake that was made. This has been also corrected in the Discussion section.

2. Fig. 1 is a graphic presentation of data from Table 2, it may be redundant. It also contains errors.  The bars showing chronic disease and infection during pregnancy do not correspond to data in Table 2.

Response 2:

According to the Reviewer’s suggestion we deleted Figure 1 from the manuscript.

I.                    Discussion:  

In a small population      of cases (only 50 mothers of children with DD were queried)  many      risk factors overlap. E.g. 50% of women had multiple gestations and nearly      50% had infections or chronic disease during pregnancy.   Were      these risk factors occurring in the same women?    If yes,      they may be linked and this would influence the interpretation of data.

Response 1:

The results regarding multiple gestation cases/controls have been corrected according to the Reviewer’s comments. 

In the modified version, the study results 12% of cases and 13.5% of controls had multiple gestations and nearly 46% of  cases and 2.9% of controls had infections during pregnancy;

6 multiple gestation cases  – all of those reported infections in pregnancy

14 multiple gestation controls – all of which did not report infections in pregnancy

This shows that these risk factors were not necessarily occurring in the same women.
